# The Inhibition of TREK-1 K^+^ Channels via Multiple Compounds Contained in the Six Kamikihito Components, Potentially Stimulating Oxytocin Neuron Pathways

**DOI:** 10.3390/ijms25094907

**Published:** 2024-04-30

**Authors:** Kanako Miyano, Miki Nonaka, Masahiro Sakamoto, Mika Murofushi, Yuki Yoshida, Kyoko Komura, Katsuya Ohbuchi, Yoshikazu Higami, Hideaki Fujii, Yasuhito Uezono

**Affiliations:** 1Department of Pain Control Research, The Jikei University School of Medicine, Tokyo 105-8461, Japan; k.miyano@jikei.ac.jp (K.M.); minonaka@jikei.ac.jp (M.N.); 3a16053@ed.tus.ac.jp (M.S.); mk06orange@gmail.com (M.M.); pp18094@st.kitasato-u.ac.jp (K.K.); 2Department of Dentistry, National Cancer Center Hospital, Tokyo 104-0045, Japan; 3Laboratory of Pharmacotherapeutics, Faculty of Pharmacy, Juntendo University, Chiba 279-0013, Japan; 4Laboratory of Medicinal Chemistry, School of Pharmacy, Kitasato University, Tokyo 108-8641, Japan; fujiih@pharm.kitasato-u.ac.jp; 5Laboratory of Molecular Pathology and Metabolic Disease, Faculty of Pharmaceutical Sciences, Tokyo University of Science, Chiba 278-8510, Japan; groadmlargo@gmail.com (Y.Y.); higami@rs.tus.ac.jp (Y.H.); 6Tsumura Research Laboratories, Tsumura & Co., Inashiki 200-1192, Japan; oobuchi_katsuya@mail.tsumura.co.jp; 7Supportive and Palliative Care Research Support Office, National Cancer Center Hospital East, Chiba 277-8577, Japan; 8Department of Comprehensive Oncology, Nagasaki University Graduate School of Biomedical Sciences, Nagasaki 852-8523, Japan

**Keywords:** Kampo, kamikihito, TREK-1, KCNK2, herbal medicine, oxytocin, paraventricular nucleus, supraoptic nucleus

## Abstract

Oxytocin, a significant pleiotropic neuropeptide, regulates psychological stress adaptation and social communication, as well as peripheral actions, such as uterine contraction and milk ejection. Recently, a Japanese Kampo medicine called Kamikihito (KKT) has been reported to stimulate oxytocin neurons to induce oxytocin secretion. Two-pore-domain potassium channels (K2P) regulate the resting potential of excitable cells, and their inhibition results in accelerated depolarization that elicits neuronal and endocrine cell activation. We assessed the effects of KKT and 14 of its components on a specific K2P, the potassium channel subfamily K member 2 (TREK-1), which is predominantly expressed in oxytocin neurons in the central nervous system (CNS). KKT inhibited the activity of TREK-1 induced via the channel activator ML335. Six of the 14 components of KKT inhibited TREK-1 activity. Additionally, we identified that 22 of the 41 compounds in the six components exhibited TREK-1 inhibitory effects. In summary, several compounds included in KKT partially activated oxytocin neurons by inhibiting TREK-1. The pharmacological effects of KKT, including antistress effects, may be partially mediated through the oxytocin pathway.

## 1. Introduction

Oxytocin, a neuropeptide composed of nine amino acids, was discovered in 1906 as a hormone that facilitates uterine contractions [1]. Oxytocin is secreted from the posterior pituitary following synthesis in the hypothalamic paraventricular nucleus (PVN) and supraoptic nucleus (SON) [2]. Studies have revealed that oxytocin has diverse physiological effects, such as the formation of maternal and social behaviors, anti-stress effects, and the suppression of food intake [3].

The herbal medicine Kamikihito (KKT), composed of 14 components (Table 1), is prescribed in Japan for neurosis, mental anxiety, insomnia, and anemia, and it exhibits effects similar to oxytocin [4,5]. Recent animal studies show that KKT stimulates oxytocin secretion, suggesting that some KKT effects may be mediated via oxytocin signaling [5]. Oxytocin released from the PVN or SON in the hypothalamus mediates central and peripheral actions [6]. Recently, potassium channel subfamily K members 2 (KCNK2), 3, and 9, members of the two-pore-domain K^+^ channel (K2P) family that regulate resting membrane potential, have been found to be expressed in both the PVN and SON [7,8]. Among these channels, TREK-1 (KNCK2) can be modulated via both chemical and mechanical stimuli to regulate membrane potential and depolarization, resulting in neuronal excitation [9,10,11,12]. Because KKT elicits oxytocin secretion [4,5], it is hypothesized that the pharmacological effects of KKT, such as anxiolytic and antipsychotic effects, may result from oxytocin secretion by inhibiting TREK-1 released from the PVN or SON, where TREK-1 channels are extensively expressed [7]. Although TREK-1 channels are reported to be involved in the secretion of neuroendocrines [13], there are no reports regarding the involvement of TREK-1 channels on the secretion of oxytocin or the increased activities of oxytocin neurons. We, therefore, assessed the effects of 14 KKT components and their compounds on the inhibition of TREK-1 channels to elucidate their role in oxytocin secretion and subsequent signaling.

As a result, we demonstrate in the present study that six of the fourteen KKT components inhibited the TREK-1 activity. Moreover, 22 of the 41 compounds included in the six components exhibited TREK-1 inhibitory properties. Numerous compounds within KKT, may activate neurons such as oxytocin neurons, by inhibiting TREK-1 activity.

## 2. Results

### 2.1. KKT Inhibited TREK-1 Activity Induced via a TREK-1 Agonist, ML335, in a Dose-Dependent Manner

We assessed the effects of varying KKT concentrations (1–300 µg/mL) on TREK-1 activity in human TREK-1-expressing cells using the FluxOR^TM^ potassium ion channel assay [14]. ML335 at 10^−5^ M was used as a selective TREK-1 agonist, and its EC_50_ was reported to be 14.3 ± 2.7 µM [15]. First, the cells stably expressing TREK-1 or not were treated with either the vehicle or ML335. Appendix A illustrates that ML335 failed to increase the intensity of fluorescence in non-transfected cells. On the other hand, in TREK-1-expressing cells, ML335 increased the intensity of fluorescence compared to the vehicle (Appendix A). TREK-1 expressing cells were next pretreated with the vehicle for 2 min and subsequently treated with either the vehicle or ML335. Appendix A shows that ML335 significantly activated TREK-1 compared to the vehicle in TREK-1-expressing cells. Therefore, the effects of KKT were analyzed using these experimental conditions. The cells were pretreated with either the vehicle or KKT (1–300 µg/mL) for 2 min and subsequently treated with the TREK-1 agonist ML335 [15]. Figure 1A illustrates that pre-treatment with KKT by itself had almost no effect on TREK-1, similar to the vehicle pre-treatment. ML335-induced TREK-1 activity was lower in cells pretreated with KKT than in cells pretreated with the vehicle (Figure 1A). TREK-1 activity was estimated using the time to peak fluorescence following ML335 treatment. The enhancement in the time to peak values meant the suppression of TREK-1 activity, as shown in Figure 1A. KKT inhibited the ML335-induced TREK-1 activity in a dose-dependent manner (Figure 1A,B). Specifically, KKT > 100 µg/mL significantly reduced ML335-induced TREK-1 activation (Figure 1B).

### 2.2. Six of the Fourteen Herbal Components Contained in KKT Were Involved in TREK-1 Inhibition

To identify which medicinal herbs contained in KKT were responsible for the inhibitory effects of TREK-1 activity (Table 1), we subsequently analyzed the effects of individual components (10 µg/mL) composed of KKT. Figure 2 illustrates that *Poria* significantly suppressed ML335-induced TREK-1 activation. Additionally, five other ingredients, *Longan arillus*, *Bupleuri radix*, *Ziziphi semen*, *Zingiberis rhizoma*, and *Angelicae acutilobae radix*, significantly inhibited ML335-induced TREK-1 activation (Figure 2). In contrast, *Glycyrrhizae radix*, *Saussureae radix*, *Ginseng radix*, *Polygalae radix*, *Astragali radix, Gardeniae fructus*, *Atractylodis lanceae rhizoma*, and *Ziziphi fructus* failed to activate ML335-induced TREK-1 activities (Figure 2). Therefore, we analyzed *Poria*, *Longan arillus*, *Bupleuri radix*, *Ziziphi semen*, *Zingiberis rhizoma*, and *Angelicae acutilobae radix* shown in the blue column of Figure 2 in subsequent experiments. 

Subsequently, we assessed the dose-dependent effects of the six identified active components on ML335-induced TREK-1 activation. *Poria, Zingiberis rhizoma*, and *Angelicae acutilobae radix* suppressed ML335-induced TREK-1 activation in a dose-dependent manner (Figure 3A,E,F). In contrast, *Longan arillus*, *Bupleuri radix*, and *Ziziphi semen* exhibited significant inhibitory effects on TREK-1 activity at 3 µg/mL compared with 10 µg/mL (Figure 3B–D). These results indicated that a minimum of 3 µg/mL of the six identified components was required to significantly suppress ML335-induced TREK-1 activity. 

### 2.3. Specific Compounds Contained in Six of the Fourteen Herbal Components of KKT Were Involved in TREK-1 Inhibition

To confirm which compounds among the six identified components inhibit TREK-1 activity, we screened their representative compounds at a concentration of 10^−5^ using the FluxOR^TM^ potassium ion channel assay. Among the *Poria* components (pachymic acid, ergosterol, and adenosine), pachymic acid and adenosine significantly suppressed ML335-induced TREK-1 activation (Figure 4A). Among the *Longan arillus* components (myristic acid, myristicin, tartaric acid, corilagin, and gallic acid), myristic acid, myristicin, and gallic acid significantly inhibited ML335-induced TREK-1 activation (Figure 4B). Among the *Bupleuri radix* components (saikogenin A, saikosaponin b2, saikogenin D, saikosaponin C, and α-spinasterol), saikogenin A and saikogenin D significantly suppressed ML335-induced TREK-1 activation (Figure 4C). Among the *Ziziphi semen* components (betulin, spinosine, magnoflorine, puerarin, swertisin, jujuboside A, jujuboside B, and betulinic acid), magnoflorine, puerarin, and swertisin significantly inhibited ML335-induced TREK-1 activation (Figure 4D). Among the *Zingiberis rhizoma* components (6-gingerol, 8-gingerol, 10-gingerol, 6-shogaol, 8-shogaol, 10-shogaol, zingerone, citral, paradol, and borneol), seven components (6-gingerol, 8-gingerol, 6-shogaol, 10-shogaol, citral, paradol, and borneol) significantly reduced ML335-induced TREK-1 activity (Figure 4E). Among the *Angelicae acutilobae radix* components (palmitic acid, linoleic acid, nicotinic acid, senkyunolide, senkyunolide H, levistolide A, bergapten, umbelliferone, feruloyltyramine, and xanthotoxin), five components (linoleic acid, senkyunolide H, levistolide A, bergapten, and xanthotoxin) significantly inhibited ML335-induced TREK-1 activity (Figure 4F). In summary, 22 of 41 compounds in the six KKT components inhibited TREK-1 activity.

## 3. Discussion

In this study, we demonstrated that a Japanese herbal Kampo medicine, KKT, inhibited TREK-1 channel activity elicited via the TREK-1 activator ML335. Additionally, we demonstrated that six of the fourteen components of KKT exhibited inhibitory effects on TREK-1 activity. Furthermore, 22 of the 41 compounds contained in the six components inhibited TREK-1 activity. This indicated that the 22 compounds in KKT exhibit the potential to inhibit TREK-1 channels.

TREK-1 channels, which are two-pore-domain potassium (K2P) channels, regulate the resting potential of excited cells, such as neuronal and endocrine cells [8,16]. K2P channels, including TREK-1 and 2, and TASK1-3 stabilize the resting potential of both excitable and unexcitable cells. The inhibition of K2P channels depolarizes membranes to elicit neuronal excitation and the secretion of neurotransmitters from neuronal cells [16,17]. Our results demonstrate that KKT inhibits TREK-1 activities, indicating that KKT may elicit neuronal and/or secretory cell activity through mechanisms that are easily elicited via certain receptor agonists on cell membranes [4]. TREK-1, TREK-2, and TASK-1 are expressed in the brain [7], specifically in the PVN and SON, where oxytocin-containing neurons are selectively expressed [7,18]. 

In experiments on the effects of KKT, it was recently reported that KKT exhibits antistress activities, potentially through oxytocin-mediated signaling, by increasing oxytocin secretion from oxytocin neurons. However, the mechanism through which KKT increases oxytocin secretion remains unclear [5]. In another study, KKT directly stimulated oxytocin neurons, which was confirmed with an in vitro electrophysiological recording assay using oxytocin neurons, likely by activating oxytocin receptors (ORs) endogenously expressed in these neurons [4]. Additionally, 3 of 14 KKT components activated ORs [4]. Among these three components, *Angelicae acutilobae radix* and *Zingiberis rhizoma* also inhibited TREK-1 activity, based on our present results, indicating that these KKT components may both activate ORs and inhibit TREK-1 to cause oxytocin secretion. Based on these findings, we propose an additional mechanism of enhanced oxytocin neuronal activity via KKT. Specifically, numerous compounds in KKT induce the suppression of TREK-1, resulting in the depolarization of oxytocin neurons and subsequent oxytocin secretion. 

TREK-1 channels are involved in numerous CNS diseases, including depression and ischemia [17]. Additionally, several studies have focused on finding inhibitors of TREK-1 to be used as therapeutic drugs for such diseases [19]. TREK-1 channels are attractive targets for pharmaceutical drug development to treat depression [16]. Efforts to identify attractive compounds for the treatment of depression have explored herbal medicines as potential therapeutic sources. In line with this theory, Hebrechter et al. assessed the effects of 158 herbal medicines on the activity of several ion channels, such as TRV1 and TREK-1, which affect neuronal excitability. Sixteen of 158 components altered TREK-1 channel activities, with eight components inhibiting TREK-1 channels and the other eight components activating them [19]. Also, the genetic and pharmacological inhibition of TREK-1 alters depression-related behaviors and neuronal plasticity in mice, indicating the significance of these channels as therapeutic targets for neuronal disorders [20]. The clinical development of TREK-1 inhibitors as therapeutic drugs, including spadin, a prospective therapeutic compound for depression [21], is currently undergoing a phase II clinical trial [17,21]. 

Our present results demonstrate that 22 of the compounds included in 6 of the 14 KKT components inhibited TREK-1 channels. Among the 22 compounds, we lacked knowledge of similarities or differences in their characteristics. However, KKT is a promising source for identifying attractive therapeutic TREK-1 inhibitors. A three-dimensional (3D) structural model of TREK-1 indicated the presence of a minimum of three binding sites that inhibit TREK channel activity [16]. These findings indicate that the 22 KKT compounds identified in this study may affect different inhibitory sites in TREK-1. Curcumin, a polyphenolic compound observed in turmeric obtained from the roots of the *Curcuma longa* plant [22], inhibits TREK-1 channels, causing the secretion of cortisol from adrenocortical cells expressing TREK-1 channels [13]. Curcumin may have numerous pharmacological effects through the enhanced secretion of cortisol or other neurotransmitters owing to TREK-1 inhibition [13]. Our present findings are also helpful in finding clinically useful compounds to enhance oxytocin signaling, as we found 22 of 41 compounds included in 6 of 14 KKT components that inhibit TREK-1 channel activities.

The limitations of this study include the inability to classify the 22 compounds in KKT as a group of TREK-1 inhibitors based on similarities in their molecular formula, such as the molecular composition of elements and/or lipophilicity. Therefore, it is necessary to precisely determine the 3D molecular structures of TREK-1 inhibitors to identify promising candidates.

Although numerous neuronal and endocrine activities are associated with the pathophysiology of depression, oxytocin may also be associated with the pharmacological effects of depression. The therapeutic effects of KKT on depression may involve oxytocin-mediated pathways [3] and TREK-1 channels. KKT resolved depressive behaviors in experimental rats with chronic restrictive stress [23]. Oxytocin regulates physiological processes and emotional states associated with coping and healing, including social behaviors [3]. Traditional herbal Kampo medicine is currently scientifically assessed with multidisciplinary and comprehensive research [24] such as “KAMPOmics” [25]. It is plausible to anticipate that certain pharmacological effects of KKT are associated with physical activities caused by oxytocin and/or TREK-1 channel inhibition.

## 4. Materials and Methods

### 4.1. Chemicals and Reagents

The following reagents were used: fetal bovine serum (FBS) and geneticin (Gibco, Carlsbad, CA, USA); a penicillin–streptomycin mixed solution and dimethyl sulfoxide (DMSO) (Nacalai Tesque, Kyoto, Japan); poly-D-lysine (PDL) (Sigma-Aldrich, Saint Louis, MO, USA); phosphate-buffered saline (PBS) (Nissui Pharmaceutical Co., Osaka, Japan); and Dulbecco’s modified Eagle’s medium (DMEM) and ScreenFect^TM^ A (Fujifilm Wako Pure Chemical, Osaka, Japan). 

KKT extract powder (Lot No. 2200137010), which is a base powder without excipients, was obtained from Tsumura & Co. (Inashiki, Japan). The formula used to produce 5 g of dried KKT extract comprised the following 14 components: *Astragali radix* (3 g, root of *Astragalus membranaceus* Bunge or *Astragalus mongholicus* Bunge), *Bupleuri radix* (3 g, root of *Bupleurum falcatum* Linné)*, Ziziphi semen* (3 g, seed of *Ziziphus jujuba* Miller var. *spinosa* Hu ex H. F. Chou), *Atractylodis lanceae rhizoma* (3 g, rhizome of *Atractylodes lancea* De Candolle or *Atractylodes chinensis* Koidzumi), *Ginseng radix* (3 g, root of *Panax ginseng* C. A. Meyer), *Poria* (3 g, sclerotium of *Wolfiporia cocos* Ryvarden et Gilbertson [*Poria cocos* Wolf]), *Longan arillus* (3 g, arillus of *Euphoria longana* Lamarck), *Polygalae radix* (2 g, root of *Polygala tenuifolia* Willdenow), *Gardeniae fructus* (2 g, fruit of *Gardenia jasminoides* Ellis), *Ziziphi fructus* (2 g, fruit of *Ziziphus jujuba* Miller var. *inermis* Rehder), *Angelicae acutilobae radix* (2 g, root of *Angelica acutiloba* Kitagawa or *Angelica acutiloba* Kitagawa var. *sugiyamae* Hikino), *Glycyrrhizae radix* (1 g, root of *Glycyrrhiza uralensis* Fischer or *Glycyrrhiza glabra* Linné), *Zingiberis rhizoma* (1 g, rhizome of *Zingiber officinale* Roscoe), and *Saussureae radix* (1 g, root of *Saussurea lappa* Clarke). A mixture of these raw materials was extracted in boiling water for 1 h, and the extract was separated from the insoluble waste. The separated extract was concentrated under reduced pressure and spray-dried to produce the KKT extract powder. The quality of KKT met the Japanese Pharmacopoeia and the standards of Tsumura & Co. The typical standards of KKT included saikosaponin b2 (0.8–3.2 mg), geniposide (27–81 mg), and glycyrrhizinic acid (6–18 mg) per 5 g of KKT. 

KKT extract powder was suspended in DMSO at 100 mg/mL, diluted 100-fold with an assay buffer, and filtered through a 0.45 μm membrane (ADVANTEC, Tokyo, Japan) to obtain a final concentration of 100 μg/mL.

### 4.2. Plasmid Constructs and Transfection 

Human TREK-1 cDNA (NM_014217) was purchased from the mammalian gene collection (accession number BC010693) and integrated into the pcDNA3.1 (+) vector (Life Technologies, Carlsbad, CA, USA). HEK-293 cells were transfected with these plasmids to generate a stable expression of the human TREK-1 complex using ScreenFect^TM^ A. To select cells expressing TREK-1, geneticin (800 μg/mL) was treated for two weeks, and clones of the cells were finally picked up. The selection of the clone was performed based on TREK-1 activity measured using the FluxOR^TM^ potassium ion channel assay. Most cells (80–90%) selected and used in the present study were activated via a TREK-1 agonist ML335 at 10^−5^ M, while HEK-293 cells that were not transfected with TREK-1 plasmids were not activated via 10^−5^ M ML335 (Appendix A).

### 4.3. Cell Culture

TREK-1-expressing HEK293 cells were cultured in DMEM supplemented with FBS (10%), penicillin (100 U/mL)/streptomycin (100 μg/mL), and geneticin (800 μg/mL) at 37 °C in a humidified atmosphere of 95% air and 5% CO_2_.

### 4.4. Measurement of TREK-1 Activity Using FluxOR^TM^ Potassium Ion Channel Assay

TREK-1 activity was assessed using the FluxOR^TM^ potassium ion channel assay kit (Invitrogen, Waltham, MA, USA), which detects potassium ion channel activity as a change in fluorescence [14], following the manufacturer’s instructions. 

The FluxOR^TM^ reagent is a fluorogenic indicator dye that was loaded into cells as a membrane-permeable acetoxymethyl (AM) ester. The FluxOR^TM^ reagent was dissolved in DMSO and further diluted with a FluxOR^TM^ assay buffer, a physiological Hank’s balanced salt solution (HBBS), for loading into cells. Loading was assisted using the proprietary PowerLoad^TM^ concentrate, a formulation of Pluronic^®^ surfactants, which acts to disperse and stabilize AM ester dyes for optimal loading in aqueous solution. Once inside the cell, the non-fluorescent AM ester form of the FluxOR^TM^ dye was cleaved via endogenous esterases into a fluorogenic thallium-sensitive indicator. The thallium-sensitive form was retained in the cytosol, and its extrusion was inhibited via water-soluble Probenecid, which blocks organic anion pumps. Before the assay, the dye-loading buffer was then replaced with a fresh, dye-free assay buffer composed of physiological HBSS containing Probenecid. During the assay, a small amount of thallium was added to the cells with a stimulus solution that opens potassium-permeant ion channels with a mild depolarization or agonist addition. Thallium was then passed into the cells through open potassium channels according to a strong inward driving force. Upon binding cytosolic thallium, the de-esterified FluxOR^TM^ dye exhibited a strong increase in fluorescence intensity at its peak emission of 525 nm. Baseline and stimulated fluorescence was monitored in real time to obtain a dynamic, functional readout of thallium redistribution across the membrane with no interference from quencher dyes. The unique FluxOR^TM^ reagent formulation allows the use of the dye in physiological saline without the need to load or assay cells in chloride-free conditions [26]. This is a major advantage over traditional approaches to thallium flux assays that utilize completely chloride-free conditions to load cells with the dye.

In the present study, the cells were seeded onto a 35 mm glass bottom dish with four compartments at a concentration of 5.0 × 10^4^ cells/0.5 mL/well (Greiner Bio-One, Upper Austria, Austria) coated with 50 µg/mL PDL. The following day, the cells were washed with PBS and treated with a fluorescent indicator at 18–24 °C for 1 h. Following a PBS wash, the loading buffer was replaced with the assay buffer. The cells were pretreated with the vehicle or KKT for 2 min at 18–24 °C. Subsequently, the cells were treated with the TREK-1 agonist ML335 [15]. The fluorescence intensity was measured every 2 s for 4 min in a real-time mode using a confocal laser scanning microscope 510 LSM (ZEISS, Land Baden-Württemberg, Germany) at excitation and emission wavelengths of 488 and 520 nm, respectively (Figure 1A). The cells were pretreated with the vehicle or KKT for 2 min at 18–24 °C in each of the wells. These cells were treated with the TREK-1 agonist ML335, and TREK-1 activity was estimated using the time to peak fluorescence. In the present study, the 10^−5^ M of ML335 (10 μM) used was based on a previous report showing that the EC_50_ of ML335 was 14.3 ± 2.7 µM in patch-clamp electrophysiology [15]. In the experimental setting, at least five cells in each well were randomly selected in a blind manner and analyzed. Experiments were performed independently at least three times.

### 4.5. Statistical Analysis

Data are presented as means ± standard errors of the means (SEMs). Statistical analyses were performed using a one-way analysis of variance (ANOVA), followed by Bonferroni’s multiple comparison test, using GraphPad Prism, version 9 (GraphPad Software, La Jolla, CA, USA). Statistical significance was set at A *p* < 0.05.

## 5. Conclusions

The effects of KKT and its 14 components on TREK-1 were assessed. KKT inhibited TREK-1 activity, and six of fourteen KKT components also inhibited TREK-1 activity. Moreover, 22 of the 41 compounds included in the six components exhibited TREK-1 inhibitory effects. Numerous compounds within KKT may activate oxytocin neurons by inhibiting TREK-1 activity.

## Figures and Tables

**Figure 1 ijms-25-04907-f001:**
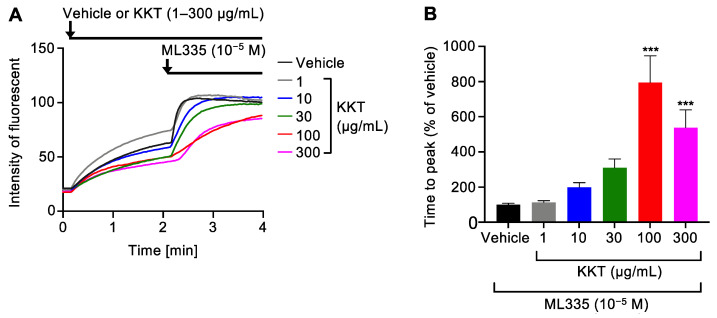
Effects of Kamikihito (KKT) on the TREK-1 activity in human TREK-1-expressing human embryonic kidney (HEK293) cells. The activation of human TREK-1 was measured using a FluxOR^TM^ potassium ion channel assay. The cells were pretreated with the vehicle or 1–300 µg/mL of KKT for 2 min and subsequently treated with a TREK-1 agonist, ML335. (**A**) Representative tracing of the mean of fluorescence intensity in randomly selected TREK-1-expressing cells (*n* = 15–30). (**B**) The activity of TREK-1 was estimated using the time to peak fluorescence following ML335 treatment (*n* = 45–90). The data are expressed as means ± standard errors of the means (SEMs). *** indicates *p* < 0.001 compared to the vehicle; Bonferroni’s multiple comparisons test followed a one-way ANOVA.

**Figure 2 ijms-25-04907-f002:**
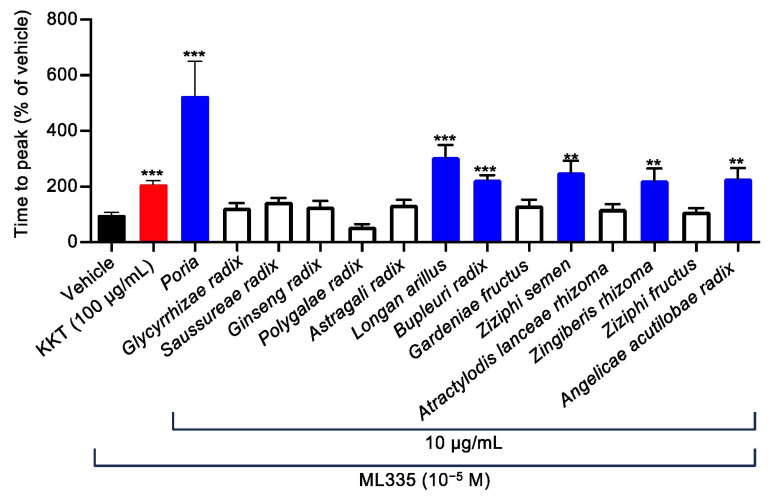
Effects of 14 crude herbal components of KKT on TREK-1 activity induced via ML335 in human TREK-1-expressing HEK293 cells. Each herbal component of KKT (10 μg/mL) or its vehicle was pretreated to the HEK293 cells, respectively. The cells were subsequently treated with the selective TREK-1 agonist, ML335 (10^−5^ M) (*n* = 15–287). The activity of TREK-1 was estimated using the time to peak fluorescence following ML335 treatment. Data are expressed as means ± standard errors of the means (SEMs). ** and *** indicate *p* < 0.01 and *p* < 0.001, respectively, compared to the vehicle; Bonferroni’s multiple comparisons test followed a one-way ANOVA.

**Figure 3 ijms-25-04907-f003:**
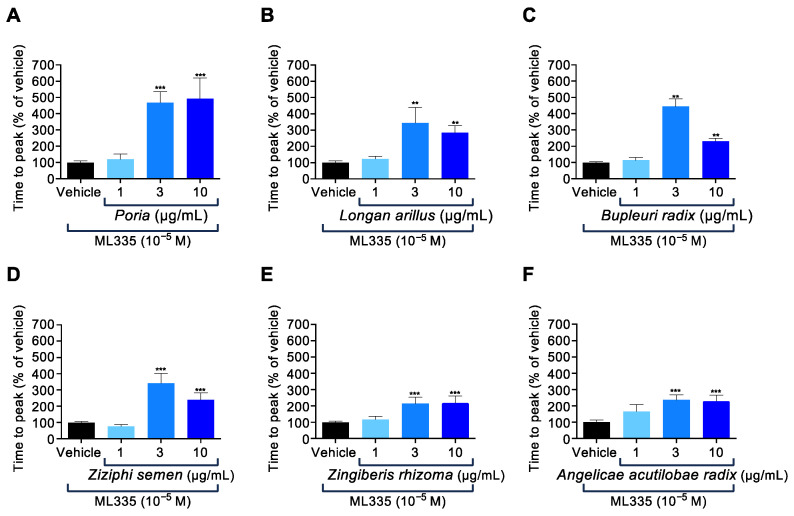
Among the 14 components in KKT, *Poria*, *Longan arillus*, *Bupleuri radix*, *Ziziphi semen Zingiberis rhizoma*, and *Angelicae acutilobae radix* inhibited ML335-induced TREK-1 activation in human TREK-1-expressing HEK293 cells. The TREK-1 expressing cells were pretreated with 1, 10, or 30 µg/mL of *Poria* ((**A**) *n* = 30–165), *Longan arillus* ((**B**) *n* = 30–240), *Bupleuri radix* ((**C**) *n* = 30–240), *Ziziphi semen* ((**D**) *n* = 30–180), *Zingiberis rhizoma* ((**E**) *n* = 30–225), or *Angelicae acutilobae radix* ((**F**) *n* = 30–180), and subsequently treated with ML335 (10^−5^ M). The activity of TREK-1 was estimated using the time to peak fluorescence following ML335 treatment. The data are expressed as means ± standard errors of the means (SEMs). ** and *** indicate *p* < 0.01 and *p* < 0.001, respectively, compared to the vehicle; Bonferroni’s multiple comparisons test followed a one-way ANOVA.

**Figure 4 ijms-25-04907-f004:**
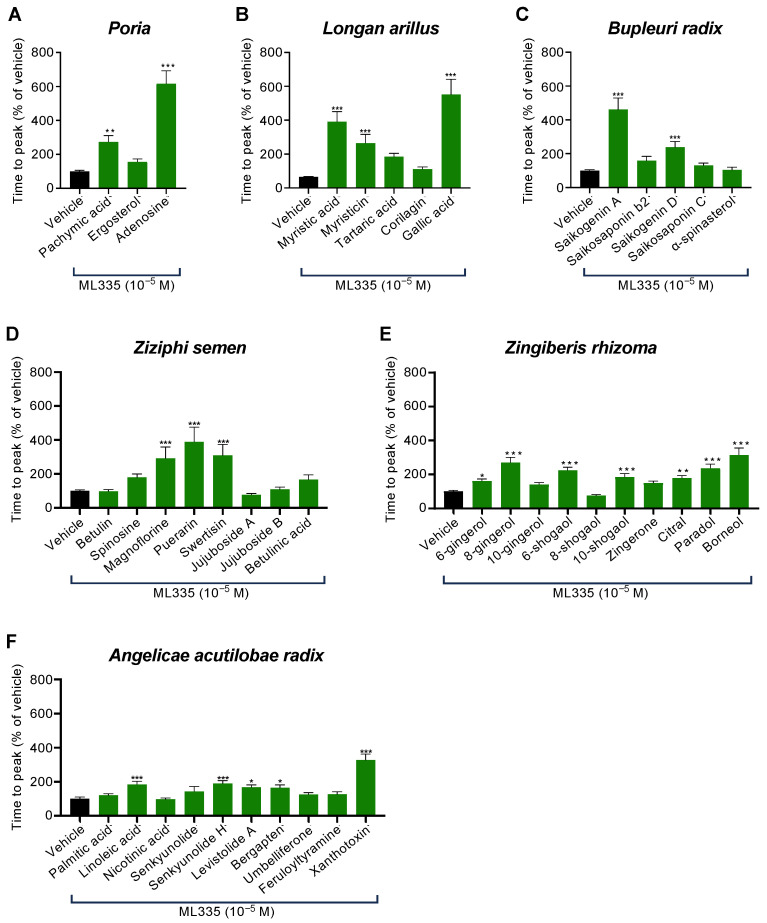
The effects of six identified active herbal components of KKT on ML335-induced TREK-1 activity in human TREK-1-expressing HEK293 cells induced via numerous compounds in components that were effective in the channels. The cells stably expressing TREK-1 were pretreated with 10^−5^ M of each component of *Poria* (pachymic acid, ergosterol, and adenosine; (**A**) *n* = 59–150), *Longan arillus* (myristic acid, myristicin, tartaric acid, corilagin, and gallic acid; (**B**) *n* = 43–180), *Bupleuri radix* (saikogenin A, saikosaponin b2, saikogenin D, saikosaponin C, and α-spinasterol; (**C**) *n* = 45–195), *Ziziphi semen* (betulin, spinosine, magnoflorine, puerarin, swertisin, jujuboside A, jujuboside B, and betulinic acid; (**D**) *n* = 40–180), *Zingiberis rhizoma* (6-gingerol, 8-gingerol, 10-gingerol, 6-shogaol, 8-shogaol, 10-shogaol, zingerone, citral, paradol, and borneol; (**E**) *n* = 45–210), or *Angelicae acutilobae radix* (palmitic acid, linoleic acid, nicotinic acid, senkyunolide, senkyunolide H, levistolide A, bergapten, umbelliferone, feruloyltyramine, and xanthotoxin; (**F**) *n* = 44–165). They were subsequently treated with ML335 (10^−5^ M), and the activity of TREK-1 was estimated using the time to peak fluorescence following ML335 treatment. The data are expressed as means ± standard errors of the means (SEMs). *, **, and ***, respectively, indicate *p* < 0.05, *p* < 0.01, and *p* < 0.001 compared to the vehicle; Bonferroni’s multiple comparisons test followed a one-way ANOVA.

**Table 1 ijms-25-04907-t001:** Components of ingredients in KKT.

Crude Drugs	Raw Materials
*Astragali radix*	Root of *Astragalus membranaceus* Bunge or *Astragalus mongholicus* Bunge
*Bupleuri radix*	Root of *Bupleurum falcatum* Linné
*Ziziphi semen*	Seed of *Ziziphus jujuba* Miller var. *spinosa* Hu ex H. F. Chou
*Atractylodis lanceae rhizoma*	Rhizome of *Atractylodes lancea* De Candolle or *Atractylodes chinensis* Koidzumi
*Ginseng radix*	Root of *Panax ginseng* C. A. Meyer
*Poria*	Sclerotium of *Wolfiporia cocos* Ryvarden et Gilbertson (*Poria cocos* Wolf)
*Longan arillus*	Arillus of *Euphoria longana* Lamarck
*Polygalae radix*	Root of *Polygala tenuifolia* Willdenow
*Gardeniae fructus*	Fruit of *Gardenia jasminoides* Ellis
*Ziziphi fructus*	Fruit of *Ziziphus jujuba* Miller var. *inermis* Rehder
*Angelicae acutilobae radix*	Root of *Angelica acutiloba* Kitagawa or *Angelica acutiloba* Kitagawa var. *sugiyamae* Hikino
*Glycyrrhizae radix*	Root of *Glycyrrhiza uralensis* Fischer or *Glycyrrhiza glabra* Linné
*Zingiberis rhizoma*	Rhizome of *Zingiber officinale* Roscoe
*Saussureae radix*	Root of *Saussurea lappa* Clarke

## Data Availability

Data are contained within the article.

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
