# Peer review of "The Inhibition of TREK-1 K+ Channels via Multiple Compounds Contained in the Six Kamikihito Components, Potentially Stimulating Oxytocin Neuron Pathways"

_ijms, 2024, doi:10.3390/ijms25094907_

Round 1

Reviewer 1 Report

Comments and Suggestions for Authors

Comments on the Quality of English Language

Author Response

Reviewer 1 comments:

 This groundbreaking study sheds light on the intricate interplay between traditional medicine components and neurological pathways, presenting promising avenues for therapeutic intervention. By exploring the inhibition of TREK-1 K+ channels by compounds found in the six Kamikihito components, the research offers novel insights into potential mechanisms underlying the modulation of oxytocin neuron pathways. However, some issues need to be addressed.

[Response]

Thank you for your valuable comments, and we answered raised issues point by point manner.

Introduction

The introduction could benefit from a stronger concluding sentence that ties the different parts together and emphasizes the research question.

[Response]

Thank you for your suggestion and we accordingly put suitable phrases in Introduction in lines 66-69 in the revised version.

  • Sentence 42: Minor redundancy - "many studies have revealed" can be replaced with"studies have revealed"

[Response]

We made correction in the revised version.

  • Sentence 43: “that oOxytocin” – please correct.

 [Response]

We corrected it.

  • Sentence 46-50 – is slightly redundant and hard to read. Please consider using a more concise form. E.g. “The herbal medicine Kamikihito (KKT), with 14 components (Table 1), is prescribed in Japan for neurosis, mental anxiety, insomnia, and anemia, and exhibits effects similar to oxytocin (4, 5). Animal studies show KKT stimulates oxytocin secretion, suggesting some KKT effects may be mediated by oxytocin signaling (5).

[Response]

As suggested, we made revision, as follows;

"The herbal medicine Kamikihito (KKT), composed of 14 components (Table 1), is prescribed in Japan for neurosis, mental anxiety, insomnia, and anemia, and exhibits effects similar to oxytocin [4, 5]. Recent animal studies show that KKT stimulates oxytocin secretion, suggesting that some KKT effects may be mediated by oxytocin signaling [5]." (lines 47-50).

Methodology

How did you select the doses for the in vitro experiments?

[Response]

We chose the concentration of ML335 (10-5 M), according to a previously reported study, which indicated EC50of ML335 as 14.3 ± 2.7 µM in a patch clamp electrophysiology with cells expressing TREK-1 [15]. Accordingly this information was added (lines 76-77 and 338-340) in the revised version.

4.2. Plasmid Constructs and Transfection

Lack of Details on Transfection Efficiency: The text mentions transfection of HEK-293 cells with plasmids to generate stable expression of human TREK-1 complex, but it does not provide details on the transfection efficiency or methods used to ensure stable integration of the plasmid into the cells. Transfection efficiency and stability are crucial for reliable results.

The methods section could include additional information such as the transfection protocol used, methods for verifying successful transfection (e.g., Western blotting), and the percentage of cells expressing the gene of interest after transfection.

[Response]

As pointed out, we totally put necessary information suggested by the reviewer, to the revised version (lines 290-296).

4.4. Measurement of TREK-1 Activity Using FluxORTM Potassium Ion Channel Assay

  • Specify the utilized control.

[Response]

First, we put principles of FluxORTM potassium ion channel assay in the revised version (lines 305-326). As for explanation of specificity of TREK-1 activities using FluxORTM potassium ion channel assay, we made experimental data and added them in Supplementary Figures 1 and 2.

  • Leaving behind KKT components could introduce variability between wells depending on how much residual KKT is present. A wash step helps to standardize the conditions for all the cells.

 It's possible the washing step is included in the manufacturer's instructions and not explicitly mentioned here. However, if not explicitly stated, adding a sentence about a wash step after KKT pre-treatment would improve the clarity and methodological rigor of the protocol.

 [Response]

As pointed out, we described detailed experimental design in the revised version (lines 333-342). Further, Figure 2 was slightly modified to indicate timing and duration of applied substances with arrow "↓".

  • Confocal microscopy settings: while the text mentions the use of a confocal laser scanning microscope for fluorescence intensity measurement, it does not specify its important parameters.

[Response]

To understand the experimental setting of confocal microscopy, we modified experimental method in the revised version (lines 333-338).

  • Minor editing is needed.

 [Response]

Thank you for the comments. As suggested, the revised manuscript was thoroughly edited by the native speaker.

Reviewer 2 Report

Comments and Suggestions for Authors

The authors of the manuscript investigated the effect of Kampo medicine Kamikihito (KKT) and its components on the ability to inhibit TREK-1 channel activity induced by ML-335 compound. Inhibition of TREK-1 channels itself is a possible mechanism explaining oxytocin receptor-induced oxytocin release observed after administration of KKT earlier. Before going to publication, I have several questions for authors.

1) How many times same time of measurement was repeated in the ion channel assay? What is a statistical unit in experiments? The authors indicated cells for each experiment. But single cell in preparation cannot be used as a statistical point.

2) What was used as a control in Ion channel assays?

3) Authors discover, what compounds of KKT have particular effect on TREK-1 channels. Can it be a potentially clinical implication, that research of these compounds alone can be more beneficial than the effects of all KKT mix altogether due to its complexity, as well as possible clinical use

4) Please check some minor misspellings. For example, “oOxytocin” in line 43

Author Response

Reviewer 2 comments:

The authors of the manuscript investigated the effect of Kampo medicine Kamikihito (KKT) and its components on the ability to inhibit TREK-1 channel activity induced by ML-335 compound. Inhibition of TREK-1 channels itself is a possible mechanism explaining oxytocin receptor-induced oxytocin release observed after administration of KKT earlier. Before going to publication, I have several questions for authors.

1) How many times same time of measurement was repeated in the ion channel assay? What is a statistical unit in experiments? The authors indicated cells for each experiment. But single cell in preparation cannot be used as a statistical point.

[Response]

As suggested, we described experimental protocols to fully explain in details in the revised version (lines 333-342).

2)  What was used as a control in Ion channel assays?

[Response]

We made a figure regarding a control in FluxORTM ion channel assay and put it in Supplementary Figures 1 and 2, which showed the comparison of data obtained from HEK293 or TREK-1-expressing HEK293 cells, and the comparison of data between vehicle and ML335 in TREK-1-expressing HEK293 cells, respectively (Supplementary Figures 1 and 2). Results are also described (lines 76-85).

3) Authors discover, what compounds of KKT have particular effect on TREK-1 channels. Can it be a potentially clinical implication, that research of these compounds alone can be more beneficial than the effects of all KKT mix altogether due to its complexity, as well as possible clinical use.

[Response]

We agree with your comments and we accordingly added some sentences regarding reviewer's comments in the revised version as follows; "Our present findings also helpful to find clinically useful compounds to enhance oxytocin signaling, as we found 22 of 41 compounds including in six out of 14 KKT components that inhibit TREK-1 channel activities." (lines 236-239).

4) Please check some minor misspellings. For example, “oOxytocin” in line 43.

[Response]

According to the comments, we thoroughly made revisions in the revised manuscript.

Reviewer 3 Report

Comments and Suggestions for Authors

Pleas improve the introduction, explaining the connection between oxytocin and TREK-1 activity. Are other neurotransmitters modulated by KKT/single compounds?

Is oxytocin secretion modulated by compounds shown to inhibit TREK-1?

Comments on the Quality of English Language Please check  the following sentence: Numerous compounds within KKT, such as oxytocin, activate neurons by inhibiting 306 TREK-1 activity. 

Author Response

Reviewer 3 comments:

Please improve the introduction, explaining the connection between oxytocin and TREK-1 activity. Are other neurotransmitters modulated by KKT/single compounds?

[Response]

Thank you for your helpful comments. According to your suggestion, we revised Introduction as follows:

"Although TREK-1 channels are reported to be involved in the secretion of neuroendocrines [13], there are no reports regarding the involvement of the channels on the secretion of oxytocin as well as the increased activities of oxytocin neurons. We therefore assessed the effects of 14 KKT components and their compounds on the inhibition of TREK-1 channels, to elucidate their role in oxytocin secretion and subsequent signaling." (lines 60-65)

Is oxytocin secretion modulated by compounds shown to inhibit TREK-1?

[Response]

As pointed out, with the previous manuscript by Maejima [4], we indicated possible involvement of three out of 14 components in KKT on the secretion of oxytocin. However, there is no report on the involvement of direct evidence of the TREK-1 channels on the oxytocin secretion in any cellular and animal experiments. Accordingly, by focusing on such three components that may cause oxytocin secretion [4], we discussed the possibility of two out of three components that may have dual effects; to activate oxytocin secretion and to inhibit TREK-1 activity (lines 203-206).

Please check the following sentence: Numerous compounds within KKT, such as oxytocin, activate neurons by inhibiting 306 TREK-1 activity. 

[Response]

As pointed out, we corrected the sentence in lines 352-353 in the revised version.

Round 2

Reviewer 1 Report

Comments and Suggestions for Authors

The authors resolved all the issues, and the manuscript was substantially improved.

Reviewer 3 Report

Comments and Suggestions for Authors

The authors reply to the reviewers comments: the paper may be accepted